# A Novel Fiber-Optic Ice Sensor to Identify Ice Types Based on Total Reflection

**DOI:** 10.3390/s23083996

**Published:** 2023-04-14

**Authors:** Chi Zhang, Chunhua Xiao, Shaorong Li, Xiaowei Guo, Qi Wang, Yizhou He, Huiyan Lv, Hongkai Yan, Dongan Liu

**Affiliations:** 1School of Optoelectronic Science and Engineering, University of Electronic Science and Technology of China, Chengdu 610054, China; 2China Aerodynamics Research and Development Center, Mianyang 621000, China

**Keywords:** ice sensor, fiber optic, icing type, ice thickness, total reflection

## Abstract

To address the issues of not accurately identifying ice types and thickness in current fiber-optic ice sensors, in this paper, we design a novel fiber-optic ice sensor based on the reflected light intensity modulation method and total reflection principle. The performance of the fiber-optic ice sensor was simulated by ray tracing. The low-temperature icing tests validated the performance of the fiber-optic ice sensor. It is shown that the ice sensor can detect different ice types and the thickness from 0.5 to 5 mm at temperatures of −5 °C, −20 °C, and −40 °C. The maximum measurement error is 0.283 mm. The proposed ice sensor provides promising applications in aircraft and wind turbine icing detection.

## 1. Introduction

Aircraft and wind turbine icing is a common natural phenomenon [1]. Aircraft icing leads to lifting reduction, drag increase, and aerodynamic performance deterioration, which affects flight safety. Ice shedding strikes engine blades, leading to engine shutdown [2,3]. Wind turbines are usually installed in cold regions and mountainous areas with rich wind resources that are easily frozen in cold climates [4,5,6]. Icing changes the aerodynamic characteristics of wind turbine blades, resulting in a 50% reduction in power generation efficiency [7,8,9,10,11]. Icing can also cause mechanical failure of wind power generation equipment [10]. Therefore, it is very important and necessary to detect the icing of major equipment such as aircraft and wind turbines.

Currently, accurate ice detection is still a significant challenge. First of all, it is hard to increase sensor measurement accuracy and range since there are many impact factors. These impact factors include the metrics, such as mean volume diameter (MVD), liquid water content (LWC), airflow speed (V), ambient pressure (P), and icing ambient temperature (T), which lead to three ice types, e.g., glazed ice, rime ice, and mixed ice, and thus affect the measurement accuracy of ice sensors seriously due to different physical characteristics of the ices such as density, dielectric constant, optical characteristics, and adhesion.

Nowadays, the most commonly used icing detection method includes the representative technologies of piezoelectric ice sensors and resonant frequency ice sensors. Piezoelectric ceramic sensors, piezoelectric lead zirconate titanate sensors, and piezoelectric film sensors are three kinds of piezoelectric ice sensors. By delivering a sinusoidal scanning voltage to the driver, the piezoelectric ceramic sensor and the piezoelectric lead zirconate titanate sensor capture the response signal and detect the icing by evaluating the wavelet packet energy of the signal [12,13]. The icing state is reflected by the vibration of the silicon diaphragm, which senses the vibration by measuring the capacitance between the coil and the diaphragm. The icing information is detected by the change in vibration frequency [14]. The resonance frequency measurement can estimate the additional mass by measuring the resonance frequency of the probe vibration. When a blade with an ice sensor on its surface is put in the icing environment, the resonant frequency will change with the increase in the mass of ice, while the resonant frequency does not change without ice mass [15]. The difference in ice dielectric constant results in different output signals of piezoelectric ice sensors at the same ice thickness, and the difference in ice density also results in large difference in the resonant frequency when the ice sensors vibrate; therefore, the ice sensor cannot accurately measure the ice thickness [16,17,18]. In addition, the extended ice sensors change the aerodynamic shape and impact the flow field as well as the aerodynamic performance of the wind turbine or aircraft [9]. Therefore, ice sensors have to be positioned on the aircraft nose and wind turbine hub, although these positions are not real icing ones.

Fiber sensor provides a solution for the ice detection problem for wind turbines and aircraft. There are some advantages of fiber-optic ice sensors including their small size and ability to be installed on the curve surface. Thus, they can be installed on the leading edge of wind turbine blades or wings but also can identify the ice type and obtain accurate icing information. Using the scattered and reflected light in ice, Aris A. Ikiades et al. proposed a method to measure the ice thickness in real time in 2007 [16]. Further, based on this concept, an array fiber-optic ice sensor with a measuring accuracy of 0.5 mm was developed. However, this array sensor was too large to be conformally installed on the curvature airfoil [17]. Due to the fiber-optic bundle’s small probe size, Ling Ye et al. put the idea of using fiber-optic bundles with an inclined angle between the end face and the axis to detect the ice thickness and ice type [18]. The ice types can be distinguished when there are some air gaps between the rime ice and the probe due to the total reflection. If there is not an existent air gap, the ice type is glazed ice. However, it is hard to guarantee that there is always an air gap between the rime ice and the probe, which reduces the ability to identify ice types.

To address the problems mentioned above, we propose a smaller fiber-optic ice sensor that can identify the ice type and measure ice thickness in the present study. The total reflection was applied in the probe structure design of the fiber-optic ice sensor. The numerical simulation and test validation were carried out to determine the light intensity and ice thickness relationship.

## 2. Mathematical Modeling of Fiber-Optic Ice Sensors

Based on the assumption of the quasi-Gaussian distribution, the ray tracing method was applied to simulate the optical field from the fiber-optic ice sensor, including the light reflection at the ice–air interface and the light scattering within the ice [19,20,21].

The fiber-optic ice sensor primarily detects the backscattered light inside the ice and reflected light at the ice–air interface as the light intensity. Equation (1) presents the received light intensity, including Equations (2) and (3), which present the mathematical models of reflection light intensity and the scattering light intensity, respectively.
(1)$$I_{REC}=I_{REF}+I_{SCA}$$
where $I_{REF}$ represents the light intensity reflected from the ice–air interface, and $I_{SCA}$ represents the light intensity backscattered within the ice. The subscripts of *REC*, *REF*, and *SCA* denote the receiving, reflecting, and scattering of light, respectively.
(2)$$I_{REF}=\frac{K_{0}\Phi_{0}}{\pi \xi^{2}\left(h\right)}\exp\left[-p^{2}/\xi^{2}\left(h\right)\right]\cdot \frac{\delta }{1+g}$$
where $K_{0}$ represents the light loss through the transmitting fiber,$\Phi_{0}$ represents the light luminous flux from the coupling of the light source and the fiber, and ξ(h) represents the equivalent radius of the quasi-Gaussian spot at the ice thickness h. p is the polar coordinate length of the light spot at h, and δ is the reflectivity of the ice–air interface. g is a constant related to the roughness of the ice surface, g∈[0:1], g=0 means that the ice surface is an ideal mirror surface, and a larger g means a rougher surface.
(3)$$I_{SCA}=I_{0}\cdot \frac{3\Gamma \lambda^{2}S\sum_{i}\frac{\left[I_{1}\left(\alpha ,n,\theta \right)+I_{2}\left(\alpha ,n,\theta \right)+I_{3}\left(\alpha ,n,\theta \right)\right]n_{r}(D_{i})}{D_{i}^{3}}\cdot \Delta D_{i}}{4\pi^{3}r^{2}\rho }$$
where $I_{0}$ is the incident light intensity, Γ is the mass density of scattered particles in the ice, and r is the distance between the microbubble and the detection surface. ρ is the relative density of the microbubble; λ is the wavelength of the incident light; S is the cross-sectional area of the ice along the light propagation direction; $I_{1}$, $I_{2}$, and $I_{3}$ are the vertical, forward, and backward components of the scattered light intensity, respectively, which are related to the diameter α, the refractive index n, and the scattering angle θ of the microbubble. $n_{r}\left(D_{i}\right)$ is the normalized function of the microbubble. D is the diameter of the microbubble.

## 3. Design of Fiber-Optic Ice Sensor

### 3.1. Structure Design

The proposed fiber-optic ice sensor is shown in Figure 1. The sensor probe consists of a prism and four fibers. The four fibers are inserted in a K9 glass. The prism material is LAK7, which is glued with the K9 glass. As a result, the probe is very small with a diameter of less than 6 mm, which can be easily mounted on the surface of an airfoil with small curvature. The four fibers include a transmitting fiber (TF) and three receiving fibers. The three receiving fibers are receiving fiber 1 (RF1), receiving fiber 2 (RF2), and receiving fiber 3 (RF3), where the end face of RF3 is oblique with an angle of 30° to the axis. RF1 and RF2 are used to detect the ice thickness. RF3 is used to determine the ice type. The light source is input into the transmitting fiber and three photoelectric conversion devices are separately put near the receiving fibers.

### 3.2. Circuit Design

The photoelectric conversion and amplifier circuit are key components of the fiber-optic ice sensor, as shown in Figure 2. For all three receiving fiber-optics, the conversion and amplifier circuits are the same. The photoelectric converter is a high-sensitivity silicon photoelectric triode with module LSSPD-1.2. The terminal of each RF is connected to the photoelectric triode. The preamplifier chip module is AD620. The converted signal is input into AD620 to obtain a differential amplification output signal. The whole circuit is powered by a DC-regulated power with module QF1722M-2. Therefore, the received light intensity is a function of the output voltage.

## 4. Simulation Results

The detection process of the sensor is modeled by the ray tracing method. The luminous power of the light source is 0.24 W (λ=980nm). The distance between the light source and the transmitting fiber is 1 mm, and the coupling efficiency of the light source and the fiber is assumed to be 100%. The parameters of the light source are shown in Table 1.

The fiber core is made of PMMA with a refractive index of 1.48 (λ=980nm), and the fiber cladding is made of PVC with a refractive index of 1.38 (λ=980nm). The structure and material parameters of the fibers are shown in Table 2 and Table 3, respectively.

The refractive index of ice is influenced by various factors, and 1.309, which is more representative, is chosen as the refractive index of ice in this model [16,22,23]. The ice–air interface has a 4% reflectivity [16]. By adding Mie scattering particles into the ice, an ideal glazed ice model and two typical rime ice models were established. By adjusting the scattering particle size (SPS) and scattering particle number density (SPND), we can figure out rime ice 1 and rime ice 2. The SPS in rime ice 1 is bigger than that in rime ice 2, and the SPND inside rime ice 1 is smaller than that inside rime ice 2. The parameters of the rime ice model are shown in Table 4 [17].

The reflection is dominantm and the scattering is tiny when the light propagates through the glazed ice. Because a large number of microbubbles stay in the rime ice, the scattering and absorption become dominant in rime ice. Therefore, the transmittance of rime ice is much lower than that of glaze ice. Figure 3 shows the ray tracing in the glazed and rime ices. In Figure 3a, at the tilted end of the discriminating fiber, all the incident lights come from the ice near the RF3 side and are reflected away because the incident angles are greater than 60°. In Figure 3b,c, the light strongly scatters in all directions in rime ice. Some lights can enter the discriminating fiber due to small incident angles, which come from the ice near the RF1 and RF2 sides.

In the glazed ice case, when the ice thickness is small, there is no overlap between the reflected light spot and the end face of the receiving fiber, as shown in Figure 4a. Therefore, the received light intensity of the receiving fiber is 0. With the increase of the ice thickness, the reflected light spot begins to overlap with the receiving end face of the receiving fiber (Figure 4b). The overlapping area of the reflection spot and the end face of the receiving fiber are defined as the effective receiving area. The effective receiving area and the received light intensity increase with the increase of the ice thickness. When the ice thickness continues to increase, the receiving end face of the receiving fiber is completely covered by the reflected light spot (Figure 4c) where the received light intensity is almost maximum. After that, the light spot continues to increase, and the effective receiving area does not increase. However, the light intensity of the effective receiving area decreases with the increase of the ice thickness. The trend is shown in Figure 5a. Because RF1 is closer to TF than RF2, the received light intensity is bigger. The simulation results can be seen in Appendix A.

In the rime ice cases shown in Figure 5b,c, the scattered light intensity depends on the ice thickness. For small ice thickness, only a part of light enters the RFs and most of the light escapes into the air. With the increase of the ice thickness, more light is scattered, and the scattering light intensity rapidly reaches maximum within the ice thickness range of 0.5–2 mm. When the ice thickness continues to increase, secondary scattering will occur and the received light intensity will increase slowly. On the other hand, the scattered light intensity also depends on SPS and SPND. Due to large SPS and small SPND in rime ice 1, the scattering light intensity is lower. Therefore, the received light intensity in rime ice 1 is lower than that in rime ice 2.

To detect the ice type, we calculated the received light intensity in RF3 as a function of ice thickness. In rime ice cases, a large number of lights enter RF3. The received light intensity in rime ice 2 is larger than that in rime ice 1, which can be used for discriminating the rime ice type. In the glazed ice case, there is no light entering RF3, which can be used for discriminating between rime ice and glazed ice.

To detect ice thickness, we calculated the light intensity modulation function, which is defined the received light intensity in RF2 divided by that in RF1, as shown in Figure 6. Figure 6a shows that the measurement range of glazed ice thickness is from 1 to 5 mm. Figure 6b,c show the measurement range of rime ice thickness is from 0.5 to 5 mm. Figure 6 shows that the light intensity modulation functions of the three types of ice show great differences, with the curve of glazed ice showing a slow growth followed by a sharp growth, and rime ice showing a sharp growth followed by a slow growth. Additionally, the sensitivity of the rime ice 2 curve is lower than that of the rime ice 1.

## 5. Experimental Detection

In our experiments, we used a chamber with brand WD6005 to obtain three low-temperatures, −5 °C, −20 °C, and −40 °C. The temperature accuracy of this chamber is within ±0.5 °C. The fiber-optic ice sensor was fixed in the acrylic plate, which is put in the chamber, as shown in Figure 7. The light source and fiber sensor structure are exactly same as the simulated ones.

To obtain the ice, compressed air is used to spray the water out from the water tank, and the water is frozen on the acrylic plate at low temperatures. To maintain the consistency of parameters, such as water droplet size, only the temperature was altered in the icing test, but the spraying parameters such as water pressure and spray angle were kept constant. The water pressure in the tank was approximately 60 Pa. The water from the pipes was quickly sprayed through the nozzles to create minute droplets with an MVD of about 20 to 40 μm. The LWC of the droplet altered with the pressure, which meant that the water in the tank suffered. When the test temperature drops to the values set above, the water droplets are sprayed on the surface of the ice sensor. Subsequently, the droplets on the probe are frozen to ice. The ice thickness is measured by a micrometer (measurement accuracy is 0.001 mm). The output voltage of the sensor is recorded by a photoelectric conversion circuit. The record interval is 30 s, until the ice thickness reaches 5 mm.

Figure 8 presents three typical ice types obtained at different temperatures. It can be seen from the glaze ice has obvious transparent characteristics and the surface is smooth, while the rime ice seems opaque, and the surface is rough. The rime ice at −20 °C shows slightly higher transparency and a relatively smooth surface compared to the rime ice at −40 °C.

Figure 9 shows the experimental correlation between the output voltage and ice thickness, which accords well with the simulated ones. For the glazed ice, RF1 and RF2 voltage output curves show a trend of increasing at the beginning and then decreasing. For the rime ice, RF1 and RF2 voltage output curves show a trend of rapid increase at the beginning and then slow their increase. The output voltage in the RF3 case differs significantly for different ice types. The experimental results can be seen in Appendix A.

Similar to the simulations, we define the light intensity modulation function (M) in which the measured output voltage for RF2 is divided by that for RF1, as shown in Figure 10. The relationship between M and h was obtained by fitting the data of the voltage and the icing temperature. The fitting polynomial equations are shown in Equation (4).
(4)$$\{\begin{array}{c} \begin{array}{c} M_{GLA}=0.31094\cdot \exp(x/3.53937)-0.36369\\ M_{RIM1}=-0.86848\cdot \exp(-x\;/\;4.73306)+0.85803 \end{array}\\ M_{RIM2}=-0.49601\cdot \exp(-x\;/\;3.68302)+0.47239 \end{array}$$
where h is the ice thickness; $M_{GLA}$, $M_{RIM1}$, and $M_{RIM2}$ are the light intensity modulation function of glazed ice, rime ice 1, and rime ice 2, respectively. The subscripts of *GLA*, *RIM*1, and *RIM*2 represent glazed ice, rime ice 1, and rime ice 2, respectively.

As mentioned above, the ice type can be achieved by using the output voltage from RF3. Then, we can detect the ice thickness by using the light intensity modulation function in each ice type case. To verify the mathematical model, we detect the ice type and thickness again, as shown in Table 5. When the stable output voltage of RF3 is around 0.0015 V, calculate the ice thickness by $M_{GLA}$; when the stable output voltage of RF3 is around 0.0023 V, calculate the ice thickness by $M_{RIM1}$; when the stable output voltage of RF3 is more than 0.0080 V, calculate the ice thickness by $M_{RIM2}$. $h_{1}$ is the ice thickness calculated from the M, and $h_{2}$ is the ice thickness measured by a micrometer. $\Delta h=\left|h_{1}-h_{2}\right|$ is the measurement’s absolute error. In the measuring range of 0.5–5 mm, the maximum measurement error of ice thickness is 0.283 mm, and the minimum one is 0.087 mm. The results demonstrate that the ice sensor can identify ice types and measure ice thickness.

## 6. Conclusions

A novel fiber-optic ice sensor was proposed based on the total reflection principle and scattering methodologies. Numerical simulation and experimental validation were performed. Some conclusions are shown as follows:(1)The ice type can be achieved by using the output voltage from RF3. Then, we can detect the ice thickness by using the light intensity modulation function in each ice type case.(2)The ice sensor can measure the glazed ice thicknesses from 1 to 5 mm. For the rime ice, the measuring range of ice thickness is from 0.5 to 5 mm.(3)The maximum and minimum ice thickness measurement errors are 0.283 mm and 0.087 mm, respectively, for the measuring range of 0.5 to 5 mm at −5 °C, −20 °C, and −40 °C.

Appendix B shows a comparison with other sensors.

## Figures and Tables

**Figure 1 sensors-23-03996-f001:**
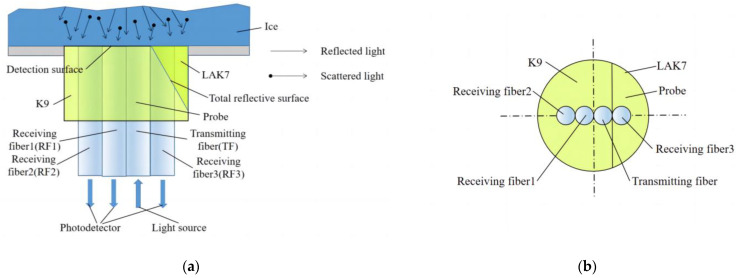
Schematic diagram of fiber-optic ice sensor structure. (**a**) Schematic diagram of the structure; (**b**) fiber arrangement diagram.

**Figure 2 sensors-23-03996-f002:**
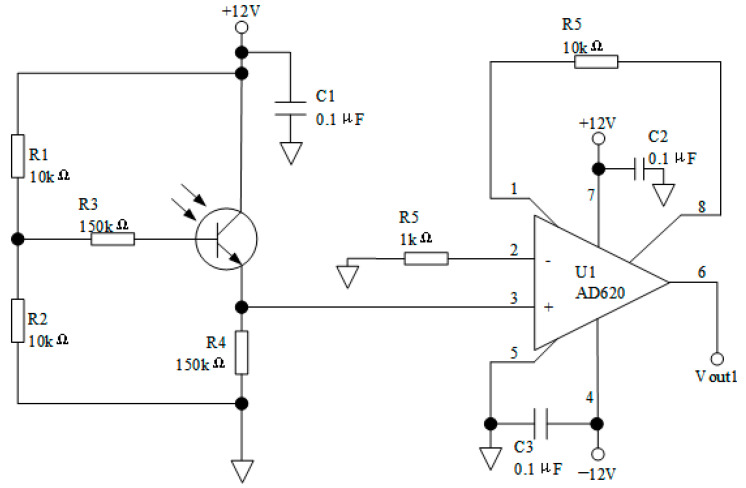
Circuit sketch of optoelectronic conversion for ice sensor.

**Figure 3 sensors-23-03996-f003:**
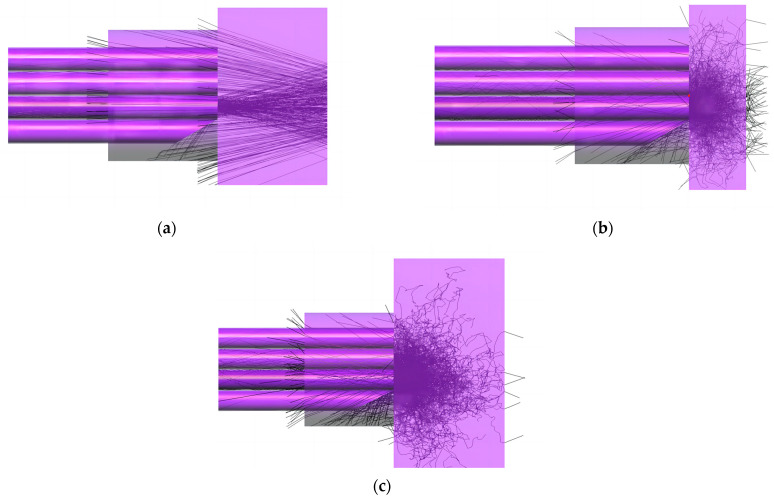
Propagation of light in different ice types: (**a**) glazed ice with 5 mm thickness; (**b**) rime ice with 2 mm thickness; (**c**) rime ice with 5 mm thickness.

**Figure 4 sensors-23-03996-f004:**
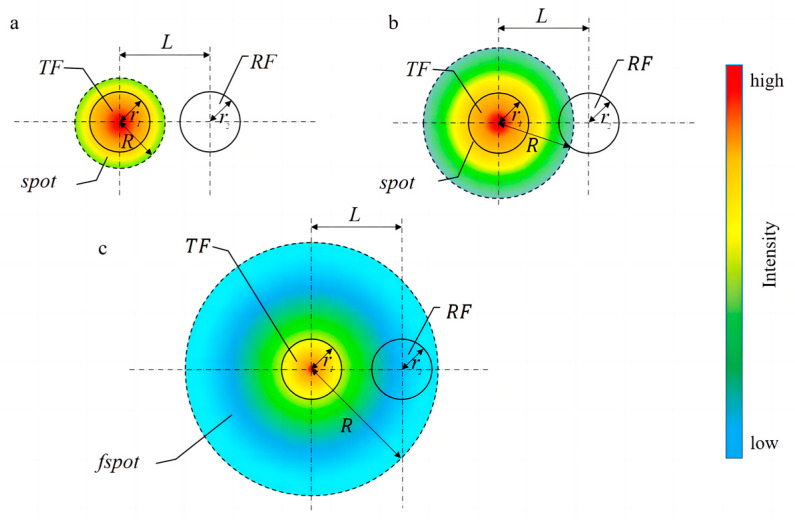
Schematic diagram of the coupling process between the reflected spot and the receiving fiber. (**a**) Reflected spot does not coincide with the receiving fiber end face; (**b**) Reflected spot partially coincides with the receiving fiber end face; (**c**) Reflected light spot completely covers the receiving fiber end face.

**Figure 5 sensors-23-03996-f005:**
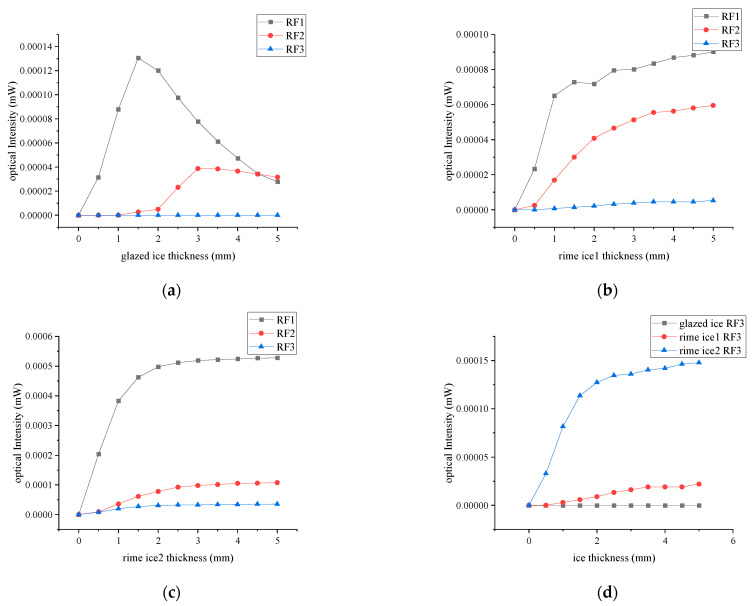
Relationship between received light intensity and ice thickness. (**a**) Received light intensity curve at glazed ice; (**b**) received light intensity curve at rime ice 1; (**c**) received light intensity curve at rime ice 2; (**d**) RF3 reception curves of different ice types.

**Figure 6 sensors-23-03996-f006:**
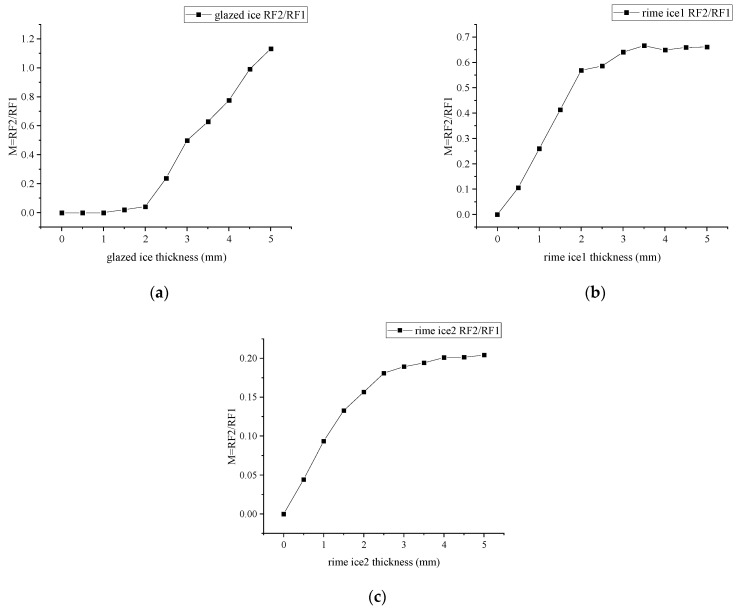
Light intensity modulation functions for different icing types: (**a**) M at glazed ice; (**b**) M at rime ice 1; (**c**) M at rime ice 2.

**Figure 7 sensors-23-03996-f007:**
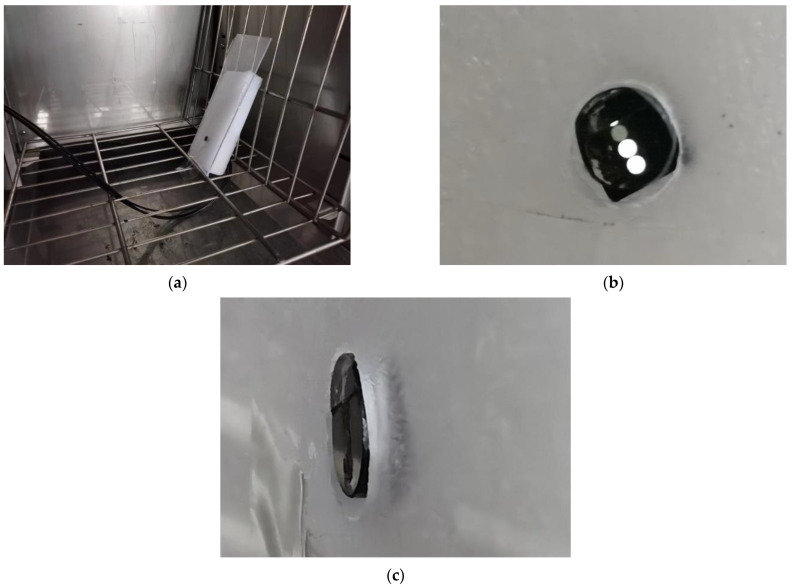
Low-temperature icing test: (**a**) fiber-optic ice sensor installation; (**b**) front view of fiber-optic ice sensor probe; (**c**) side view of fiber-optic ice sensor probe.

**Figure 8 sensors-23-03996-f008:**
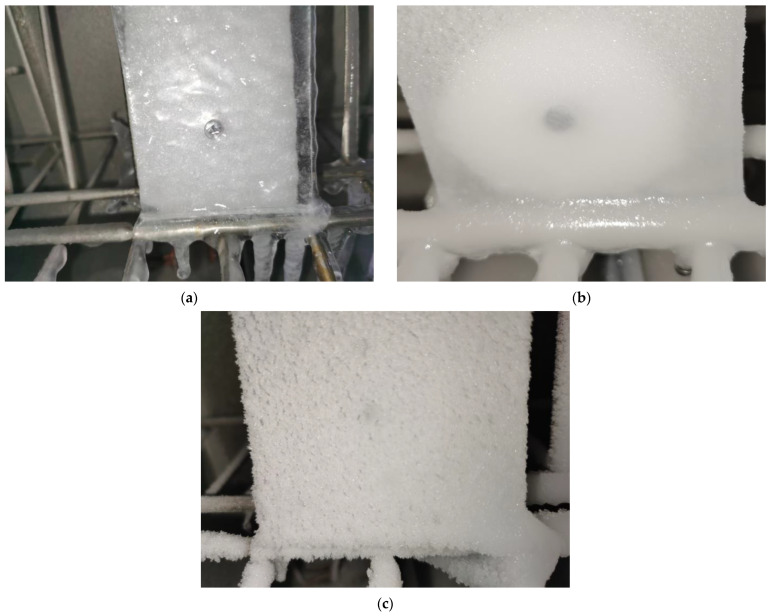
Three typical ice types for the low-temperature icing test: (**a**) glazed ice for −5 °C; (**b**) rime ice for −20 °C; (**c**) rime ice for −40 °C.

**Figure 9 sensors-23-03996-f009:**
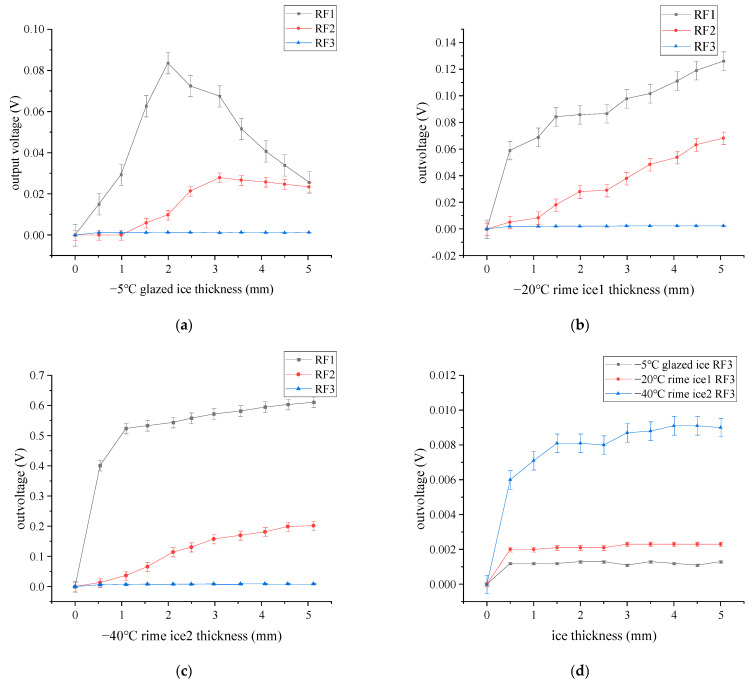
Experiment results: (**a**) glazed ice; (**b**) rime ice 1; (**c**) rime ice 2; (**d**) comparison of different ice types.

**Figure 10 sensors-23-03996-f010:**
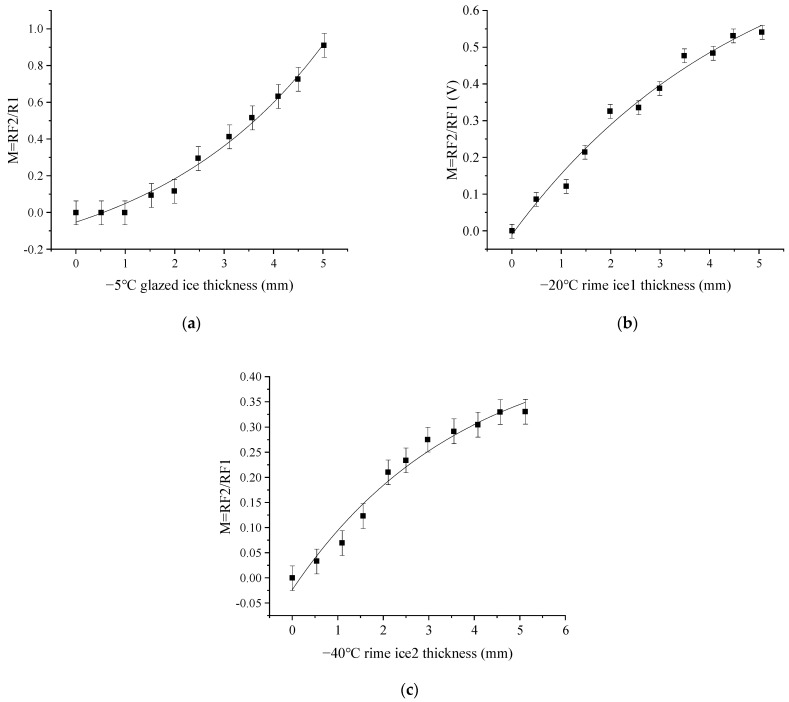
Relationship between the light intensity modulation function and ice thickness. (**a**)−5 °C glazed ice; (**b**) −20 °C rime ice 1; (**c**) −40 °C rime ice 2.

**Table 1 sensors-23-03996-t001:** Parameters of the light source.

Parameter	Numerical
Light-emitting angle	Arctan (0.5/1)
λ(nm)	980
Power (W)	0.24

**Table 2 sensors-23-03996-t002:** Structural parameters of fiber-optic.

Fiber-Optic Name	Fiber-Optic Length	Core Radius	Cladding Thickness
RF1	1000 mm	0.5 mm	0.05 mm
RF2	1000 mm	0.5 mm	0.05 mm
RF3	1000 mm	0.5 mm	0.05 mm

**Table 3 sensors-23-03996-t003:** Material properties of fiber-optic.

Structure Name	Refractive Index	Transmittance (1/m)
Core	1.48	0.9899
Cladding	1.38	0.9200

**Table 4 sensors-23-03996-t004:** Rime ice model parameters table.

Title 1	Rime Ice 1	Rime Ice 2
SPS (mm)	10, 15, 20, 25, 30	7, 8, 9, 10, 11
SPND (1/mm^3^)	1.0851 × 10^4^	1.5733 × 10^5^

**Table 5 sensors-23-03996-t005:** Test data of ice sensor validation.

M	$h_{1}(mm)$	$h_{2}(mm)$	Δh(mm)	Out Voltage (RF3) (V)	Detected Ice Type	Realistic Ice Type
0.1159	1.534	1.621	0.087	0.0015	Glazed ice	Glazed ice
0.0015	3.425	3.624	0.199	0.0015	Glazed ice	Glazed ice
0.0014	5.006	5.284	0.278	0.0014	Glazed ice	Glazed ice
0.1664	1.078	1.263	0.185	0.0023	Rime ice 1	Rime ice 1
0.3998	3.026	2.921	0.105	0.0025	Rime ice 1	Rime ice 1
0.5538	4.965	5.248	0.283	0.0025	Rime ice 1	Rime ice 1
0.0878	0.937	1.098	0.161	0.0081	Rime ice 2	Rime ice 2
0.2234	2.539	2.316	0.223	0.0091	Rime ice 2	Rime ice 2
0.3481	5.097	4.896	0.201	0.0089	Rime ice 2	Rime ice 2

## Data Availability

Data are available on request from the authors.

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
