# Peer review of "A Novel Fiber-Optic Ice Sensor to Identify Ice Types Based on Total Reflection"

_sensors, 2023, doi:10.3390/s23083996_

Round 1

Reviewer 1 Report

This works presents a proof of concept for fiber sensor intended to measure the thickness of ice and identification of type of ice. Authors perform simulations for the reflected and scattered light produced by glaze and rim ice, and how that light is detected and discriminated in the sensor based in four fibers: i) illumination fiber; ii) two receiving fiber located not equidistantly respect to illumination fiber; iii) a third receiving fiber comprising a discriminating property of scattered light based in a total reflective surface.

My first comment on this work is that the concept and design of the sensor is clever and nice. Nevertheless, as a scientific report this manuscript has many flaws and it does not reach a standard of quality. Some of the issued that the manuscript should have covered are:

1)  Eq. 1, 2 and 3 have many free parameters. The manuscript did not specify properly the range of validity and or used range of parameters in simulations.  The values assumed for these parameters were not either justified. Further, some parameter were not described as in the case of de “diameter alfa”

2) Some refractive indexes for the fiber and other materials are given, but never said at which wavelengths they correspond. 3) Figure 3 presents information and parameters that are not defined in the manuscript.

4) Simulations present unexpected variations within the trend. For instance, Figure 5b at the thickness of 2 mm shows decrease of optical density respect adjacent data points. Why?

5) The total reflection effect in fiber 3 must be true for specific wavelengths. What wvalegths were used in simulations? What type of lamp was used in experimental validation?

6) Photo in Fig 7b is out of focus.

7) As a good description of the model was not provided in the manuscript, it was difficult for this reviewer to validate the results produced by the model. Please notice that there are many free parameters such that multiple fittings with the same prediction on “h” can be obtained.

8) There are many typos in the text, more than acceptable assuming a regular reading of text prior to submission was carried out.     

9) I could not download supple. info.

I recommend the authors prepare a new version of the manuscript and resubmit it. The reviewer must have enough information to give an objective opinion of the actual scientific contribution of this work, and comment if the conclusions are really supported by data.  

Reviewer 2 Report

This paper reports a fiber-optic ice sensor fabricated based on the reflected light intensity modulation method and total reflection principle. The results are interesting, there are still some issues should be addressed.

(1) Error bars should be added to the data, such as Fig. 5, Fig. 8, Fig. 9 and table 2.

(2) The grammar should be modified carefully.  

(3) The authors should compare the performance of literature reported ice sensors and the proposed ice sensors. 

(4) The testing mechanism of the ice sensor should be better demonstrated. 

(5) The layout and form of the manuscript and figures should be moidifed。 

Reviewer 3 Report

The paper is well-drafted and organized but there are following major revisions to be done.

There are a few language errors in the whole manuscript and advised the authors to proofread the manuscript.

I advise authors to form a comparison table of performance parameters of  recent existing similar literature with the proposed work 

The resolution of all figures must be enhanced and the advised authors increase the line thickness of the figures.

Figure inset captions are very small and they should be increased

Round 2

Reviewer 1 Report

This revised manuscript has improved compared with the original version, nevertheless some more improvements are required before acceptance. For instance.

1) authors addressed the issue of describing explicitly the wavelength for the refractive indexes of pmma and pvc, authors had cited explicitly the wavelength for the refractive index of ice (in line 142 reference [17]), but in reference 17 the wavelength for the ice refractive index is at 624 nm while authors in the present manuscript present simulations and experiments at 980 nm.  

2) it is still confusing for this reviewer to interpret the importance of Eq 3. Was this equation used or not in the simulation? According to response to reviewers letter, the utilized software uses such equation to perform the ray tracing calculations. Ok, that is clear, but readers would appreciate to authors if they describe in better detail the assumptions and conditions taken in the ray tracing method. In case Eq. 3 is in fact used directly in simulations, the authors must justify the values used for each the parameters involved in such equation.  

3) there are still some typos, for instance lines 45, 87, etc.

Reviewer 3 Report

All the suggested modifications are addressed and The paper could be accepted in present form
